# Socioecological Predictors on Psychological Flourishing in the US Adolescence

**DOI:** 10.3390/ijerph17217917

**Published:** 2020-10-28

**Authors:** TaeEung Kim, Chang-Yong Jang, Minju Kim

**Affiliations:** 1Department of Epidemiology, University of California, Irvine, CA 92697, USA; ktang7711@gmail.com; 2Korea Institute of Sport Science, Seoul 01794, Korea; 3Department of Dance, Hanyang University, Seoul 04763, Korea; amy1206@daum.net

**Keywords:** flourishing, well-being, socioecological factors, quality of life

## Abstract

This study examined the relationship between children’s flourishing and socioecological factors, including individual and family characteristics. A total of 45,309 children were drawn from the 2011–2012 National Survey of Children’s Health in the US (mean age = 13.6 years; male: 51.28%). An ordinary least square (OLS) regression was performed to examine the association between socioecological variables and flourishing. This study assessed children’s individual needs (such as health, education, and emotional and behavioral development), their parents’ parenting capacity (the ability to provide basic care and safety), and family factors (presence of community resources and family income). Children’s individual characteristics, parent’s capacities, and family functions were found to be significantly associated with children’s flourishing. In conclusion, multilevel socioecological factors appeared to be associated with children’s flourishing. Hence, parents’ involvement in their children’s physical activities, as well as family and social support, is crucial for children’s flourishing. This study makes a significant contribution to the literature as research is lacking a correlation between socioecological factors and children’s flourishing; in particular, very few studies have explored or investigated the manner by which children’s socioecological health indicators and factors are associated with their flourishing.

## 1. Introduction

Flourishing, as a new definition and expression for happiness and well-being has been proposed [1,2]. Individuals can flourish by fostering positive emotions, relationships, engagement, accomplishments, meaning and purpose, or participating in voluntary community services, helping them overcome their hardships, adversities, and/or trauma. Therefore, the concept of flourishing relates to a combination of emotional happiness and self-achievement in one’s environment. In other words, flourishing relates to obtaining a sense of meaning and feeling of self-fulfillment in one’s life. Individuals can flourish through the pursuit of intrinsically satisfying goals, such as by immersing themselves in activities, relationships, and overcoming adversity; for example, by serving in one’s community.

Flourishing can help children by promoting and improving health and well-being, both emotionally and physically, in later life [3,4]. This is mainly achieved through the enhancing of the children’s resilience and creativity [5]. A group of adults was examined using longitudinal data and reported a substantial and pervasive difference in health and well-being between those who had flourished at secondary school and those who had not [4]. Moreover, they also indicated that the adults’ attained better life quality dependent on their school engagement and academic accomplishment, which again differed between those who had flourished and those who had not [4]. Flourishing is also proposed as an effective method of improving and developing robust and stable mental health in children (e.g., coping with adversity), and reported that to ensure children have the best opportunities to flourish, appropriate environments must be provided [3].

A small number of studies have investigated and reported on how healthy behaviors, such as yoga practice [6] and eating fruits and vegetables [7], influence flourishing and the health-related quality of life of adults and young adults. For example, yoga practice programs affect one’s general health characteristics and quality of life, and it is reported that those who participate in such activities have better mental and physical health benefits than their counterparts, such as improved sleep quality, less fatigue, better social support, lower weight, smoke less, higher fruit and vegetable consumption, and better mental health or flourishing [6]. Further, the participants believed yoga practices helped increase energy and happiness, and improved social relationships, sleep, and weight. Another study focused on how the consumption of fruits and vegetables is related to greater flourishing in daily life among young adults [7]. Thus, eating more fruits and vegetables could be associated with well-being, such as happiness and life satisfaction, and is also linked to higher well-being, increased curiosity, and greater creativity. However, none of the studies illustrated how participants’ healthy indicators (e.g., weight and stress levels) or socioecological factors could influence their flourishing.

As is well known, children are nested within parent and family classes, but further, they are also nested in the school class, and schools are nested within either school districts or communities. Therefore, children develop their personalities and characteristics under the influence of their parent(s), family, and multifaceted, structured society. Moreover, children’s self-judgment is unstable and incomplete, resulting from complexities in regard to their uncertain behaviors and thoughts, their parents’ lifestyles and actions, and access to safe societal infrastructures, all of which play an important role in creating and developing physical, mental health, and positive personality of children. Considering this, a multidimensional research approach is required to investigate aspects that have significant implications for this hierarchical structure [8,9].

However, research on the socioecological factors that may influence children’s flourishing is lacking, and few studies have explored how children’s socioecological health indicators are associated with flourishing. The purpose of this research, therefore, was to examine, in a large nonclinical setting, the sociological factors that affect children’s flourishing. Further, we also wished to obtain informed outcomes and increase awareness of the relationships between socioecological factors and flourishing. This would facilitate improvements in the understanding of individual and family factors that influence children’s flourishing mechanisms, which in turn could lead to being useful and helpful for preparing tailored health interventions or policies regarding the characteristics and features of the children.

For this study’s conceptual basis, we chose to apply the modified socioecological model framework [10] (see Figure 1), which focuses on the key transitions in children’s lives. Since children are nested within family classes, which are, in turn, nested within and school classes, and schools are nested within either school districts or communities, a multiple-level research approach was needed to investigate aspects that have significant implications for this hierarchical structure [10].

Such an approach can examine the major needs of children, parents, family, and community members in relation to childhood development. Thus, in this study, this model provided a systematic method of analyzing, understanding, and recording the situation experienced by children and youths within their families and the communities in which they live. Specifically, the development of the assessment framework consisted of three main parts: (1) the child’s individual needs (i.e., health, education, emotional and behavioral development, identity, family and social relationships, social presentation, and self-care skills); (2) parenting capacity (i.e., providing basic care, safety, emotional warmth, stimulation, guidance, boundaries, and stability); and (3) family factors (i.e., the family’s social integration, income, size, history, and function; parents’ employment, and housing). Sociological factors related to flourishing: family and parents’ health behaviors, including alcohol intake, smoking, family activity, and social volunteer service.

The objectives of this study were to examine the relationship between children’s flourishing and multi-sectional factors, including children’s lifestyles, family, and environments. The research hypotheses were as follows:

Overall hypothesis: There are associations between children’s socioecological factors and their flourishing.

**Hypothesis** **1.**
*Individual factors (overweight, missed school days, level of physical inactivity, sedentary lifestyle, etc.) are negatively associated with children’s flourishing.*


**Hypothesis** **2.**
*Parenting capacity (parent’s mental health, parent and child relationship, parents’ stress coping, parents’ smoking, etc.) are positively associated with children’s flourishing.*


**Hypothesis** **3.**
*Family factors (poverty, family activities, and single parent, etc.) are positively associated with children’s flourishing.*


## 2. Methods

### 2.1. Data Sampling and Study Population

A total of 45,309 children (representing 30,965,078 at the population level) were analyzed (mean age: 13.63 years, SD: 2.35; male: 51.28%). The data, as a secondary source, were extracted from the 2011–2012 National Survey of Children’s Health (NSCH) in the US. For the NSCH, random sampling and telephone household interviews were conducted by the National Center for Health Statistics of the Centers for Disease Control and Prevention [11]. The NSCH is weighted to represent the national population of non-institutionalized children aged 0–17, and provided information on a variety of physical, emotional, and behavioral health indicators relating to children, their families, and neighborhoods.

Based on a cross-sectional design, for this study, we extracted data from the 2011 and 2012 NSCH samples for children aged 10–17 because we were unable to obtain status information for respondents younger than 10. The Institutional Review Board at the authors’ institution approved this study.

### 2.2. Measures

In the NSCH, the questions relating to flourishing were rated, on average, using a five-point Likert scale ranging from five (“strongly agree”) to one (“strongly disagree”). Examples of questions are as follows: (a) “(He/She) finishes the tasks (he/she) starts and follows through with what (he/she) says (he will/she will) do”; (b) “(He/She) stays calm and in control when faced with a challenge”; and (c) “(He/She) shows interest and curiosity in learning new things.” For the current study, the scale’s internal consistency was determined to be adequate (α = 0.75).

First, we determined the children’s individual characteristics, including sex, race/ethnicity, age, days of school missed, participation in after-school activities, quality of sleep, physical activity, and sedentary behavior. Race/ethnicity was categorized into four groups: Hispanic, non-Hispanic white, non-Hispanic black, and other. The available age range was from 10 to 17 years. A composite measure for after-school activity was computed using three binary scales, including participation in any clubs/organizations, taking sports classes or being on a sports team, and participation in any other organized activity during the previous 12 months. Physical activity was measured by determining the number of days the children had exercised, played a sport, or participated in vigorous physical activity for at least 20 min. Additionally, a composite scale score for sedentary behavior was used, and this was calculated as the cumulative length of time, in hours, spent engaging in activities such as watching TV, videos, DVDs, or playing video games.

Parenting capacity concerned parents’ mental health, stress coping, smoking, and alcohol and drug use. In particular, for each child, parent’s health was classified as the average of the mother’s and the father’s physical and mental health. Parents’ stress-coping was assessed using a four-point Likert scale, and this included topics such as how well they felt they were coping with the daily demands relating to raising their children. Meanwhile, two binary items were used to measure parental health behaviors: smoking, alcohol, and drug use. Lastly, the parent and child relationship was assessed using a four-point Likert scale and an item asking how well parents share ideas or discuss things that really matter.

Family function consisted of family activity, federal poverty level (FPL), and the single parent variable. Family activity was assessed using a two-scale composite containing: (1) how often children attended a religious service, and (2) the number of days during the past week all family members had eaten together. Finally, the participants’ levels of poverty were categorized into four groups based on FPL (0–99% FPL; 100–199% FPL; 200–399% FLP; and over 400% FPL) [12]. Initially, there were nine categories relating to combined family structures and the marital/cohabitation status of each child’s parents, but we restructured these groups into a binary variable, such as single-parent vs. another family structure; this was because we believed that the health status of children with a single parent might differ from those who had a different family structure.

### 2.3. Analysis

In order to describe the study sample’s demographic characteristics, such as age, gender, race, and overweight status, unweighted and weighted descriptive statistics of the population were reported in terms of means, standard deviations, or percentages. Further, Pearson χ^2^ tests and *t*-tests with weighted counts and column percentages were performed for descriptive statistics. Moreover, an ordinary least regression was performed, and sequential regressions were also undertaken to examine the associations between socio-demographic (e.g., family activity and parents’ health behaviors) variables and flourishing. All statistical analyses were conducted using Stata^®^ 15.1 (Stata Corporation, College Station, TX, USA), and statistical tests were conducted using a 0.05 alpha level and a 95% confidence interval.

## 3. Results

The unweighted (e.g., number of participants) and weighted (e.g., mean, SD, and percentage) descriptive statistics of the study population are shown in Table 1. As can be seen in this table, the participants’ races were quite diverse, with most being non-Hispanic whites (53.89%), followed by Hispanic (20.77%) and non-Hispanic black 14.00%. Their income levels were also diverse (see Table 1). For example, 30.13% reported an FPL level of over 400%; this was followed by 200–399% FPL (28.88%), 100–199% FPL (21.12%), and 0–99% FPL (19.87%).

After testing the correlation coefficient between variables, multicollinearity was determined not to be an issue (Tolerance value for all >0.1; variance inflation factor (VIF) value <10) [13]. Further, all assumptions of ordinary least squares regression, such as normality, linearity, homoscedasticity, and independent random errors, were met. As shown in Table 2, based on our conceptual framework, all sequential regression models with covariates (e.g., individual, parent, and family factors) for flourishing were significant (*p* < 0.01). Overall, the final model was significant (*F* (32, 89341) = 82.37, *p* < 0.01), and accounted for approximately 30% (R^2^ = 0.30) of the variance in the children’s flourishing; collectively, 11 variables relating to socioecological factors significantly predicted flourishing.

In terms of the children’s individual needs, seven predictors were found to be statically significant: gender (*β* = 0.17, *p* < 0.01); race/ethnicity—Hispanic (*β* = 0.09, *p* < 0.01) and others (*β* = 0.04, *p* < 0.01); after-school activity (*β* = 0.09, *p* < 0.01); physical activity (*β* = 0.02, *p* < 0.01); and sedentary behavior (*β* = −0.01, *p* < 0.01). Regarding parenting capacity resources, four were found to be statistically significant: parents’ health (*β* = 0.02, *p* < 0.01); parents’ stress coping (*β* = 0.19, *p* < 0.01); and parents’ alcohol/drug use (*β* = −0.15, *p* < 0.01). In terms of family function, two were statistically significant: family activity (*β* = 0.02, *p* < 0.01) and parent and child relationship (*β* = 0.30, *p* < 0.01).

## 4. Discussion

This study examined the associations between socioecological variables (e.g., family activity, parents’ health behavior, type of family structure) and children’s flourishing, considering age, gender, and ethnicity differences. Our main concerns with regard to this study were determining appropriate methods, based on our conceptual research framework, of examining how parents and family factors influence the development of children’s flourishing [10]. Children lack autonomy, self- and social-motivation, and are socially and emotionally unstable; this means that parents, family, and environments play an important role in their development and growth [10]. Although this is the first study to examine the flourishing of children, the identification of significant socioecological factors affecting children’s flourishing can be supported by examining findings from previous research relating to children’s development and well-being; this is due to the similar characteristics between the fields, especially in regard to psychological properties.

As hypothesized, we found significant associations between children’s socioecological factors and their flourishing. In particular, parenting capacity (e.g., parents’ health, stress coping, and alcohol/drug use) and family factors (e.g., family activities, and parent and child relationship) were determined to be highly associated with children’s flourishing. In terms of parenting capacity, having a parent with good physical and mental health, who has good stress-coping skills was found to have a positive impact on children’s flourishing. In their study of Greek children aged 11–18 years [14], children’s health-related quality of life, including physical and psychological well-being, moods and emotions, and parent-child relationships, was significantly associated with parental subjective mental health status; the researchers also found that parental subjective physical health status, in turn, was strongly correlated with positive self-perception.

Our findings also revealed that negative parental health behaviors, including smoking and consumption of drugs/alcohol, showed negative influences on children’s flourishing; however, the effect of parents’ smoking was not statistically significant in this particular study. In fact, parental smoking seems to have a substantial impact on children’s health, such as causing respiratory symptoms and/or asthma [15] and increased allergic sensitization [16], rather than on their emotional and psychological development. On the other hand, parents’ consumption of alcohol and drugs was determined to be highly associated with children’s well-being and protection [17] and their personality (e.g., greater negative emotionality, aggression, stress reaction) [18]. Thus, the above studies provide some evidence that parenting capacity plays an important role in forming children’s flourishing.

Another important socioecological perspective in children’s flourishing is the role of the family. Children who engage in more family activities (e.g., attending religious services and eating meals together) and who have better relationships with their parents were found to have better flourishing. Specifically, a more favorable and happier family environment seems to cause positive flourishing in children because the children engage in more meaningful activities, such as parent-child communication, family connectedness, and interaction with family members. Moreover, family religious activity [19,20] and family time [21,22] have been found to be highly associated with emotional/psychological well-being in youths and children, respectively. Flourishing is a brand-new concept of one’s happiness and well-being [1,2]. In addition to the important role family activity plays in forming children’s flourishing, a child can also flourish through the fostering of positive relationships with parents or by participating in voluntary community services, which can help them overcome their hardships, adversities, and trauma. Children in a low-conflict, married families who are close to both parents, have the highest level of subjective well-being; further [23], in addition, the positive association between parent-child communication and children’s life satisfaction is also illustrated [24]. Loving and gentle parenthood, therefore, plays a very important role in the development of children.

Our study also uncovered other meaningful findings; for example, how several individual factors affect children’s flourishing, including being overweight, physical inactivity, and possessing a sedentary lifestyle. As shown in Table 2, being physically active, such as through after-school activities and the number of days per week physical activity is engaged in were significantly positively associated with children’s flourishing, whereas a sedentary lifestyle had the opposite effect. Practicing yoga influenced flourishing and improved the health-related quality of life of adults by providing mental and physical health benefits, including improved sleep quality, reducing fatigue, and increasing social support [6]. Further, in the above study, the participants reported that yoga helped them to improve their energy, happiness, and social relationships. Additionally, we found that children who frequently missed days of school were prone to less flourishing. This is because schools are one of the most suitable infrastructures for ensuring that the next generation develops wellness; for school-aged adolescents, schools provide many opportunities to learn and to practice diverse, healthy behaviors and academic subjects. Students do not only learn through lessons but also involve themselves in social and emotional relationships with peers and teachers [25].

In this age range, female children are more likely to flourish than male children. This is because male children’s development process is more likely to occur more slowly than that of females. Interestingly, we found race/ethnicity differences, specifically in regard to the flourishing of Hispanic children and other minors: Hispanic children were more likely to flourish than non-Hispanic white children. This may be partly because, in a society that mainly consists of Hispanic children, but also other minors, people of the same ethnicity are more inclined to congregate and feel more closeness with each other, which may provide them with more opportunities to interact, communicate, and share their ideas and thoughts with each other, which, in turn, facilitates the building of mature personality and development.

To promote and improve health and well-being, both emotionally and physically in later life, children must flourish [3,4]; this, in turn, leads to enhanced resilience and greater creativity [5]. Moreover, flourishing is also proposed as an effective method of enhancing and developing a robust and stable state of mental health (e.g., coping with adversity), as it provides appropriate environments for children [3]. Thus, the above shows that socioecological factors, from individuals and parents to family and environments, influence flourishing in a multidimensional way. In particular, children’s flourishing is affected not only by individual factors, but also by support from parent(s), family, and community, while parents’ involvement in children’s physical activities and obtaining family and social support are crucial in children’s flourishing.

This research can facilitate an enhanced understanding of individual and environmental connections with children’s flourishing mechanisms, which could, in turn, lead to public health prevention and intervention strategies that develop children’s personalities and improve their quality of life. The rationale for this research is that once it is known how parents, family, schools, and communities play important roles in children’s flourishing patterns.

This study should be interpreted in light of the following limitations: Firstly, to assess all study variables, this study utilized self-reported measurements based on respondents’ one-day recollection, which may be susceptible to recall bias, respondent bias, and/or interview bias. Secondly, the small sample size (e.g., Hispanic and others) may mean that these results are not generalizable to those living in more diverse populated areas. A future study that features larger and more diverse participants would provide better results. Thirdly, the dataset excludes households without telephones, which may result in a biased survey population due to the underrepresentation of certain segments of the population.

Despite the above limitations, we believe that they do not outweigh the contribution of this study. Nevertheless, we suggest that future studies also examine if multilevel effects can contribute to children’s flourishing status. This study revealed different dimensions of how sociological factors influence children’s flourishing in the general child population. Children’s flourishing was affected not only by individual factors, but also by family and community support. Parents’ involvement in children’s physical activities and obtaining family and social support are crucial in helping children flourish. Consequently, more attention should be paid to providing children with the necessary family and social support to help them maintain emotional health and well-being. Considering the above, we suggest that health educators and professionals, policymakers, and stakeholders consider adopting a multidimensional approach to developing intervention programs that can improve children’s quality of life.

## Figures and Tables

**Figure 1 ijerph-17-07917-f001:**
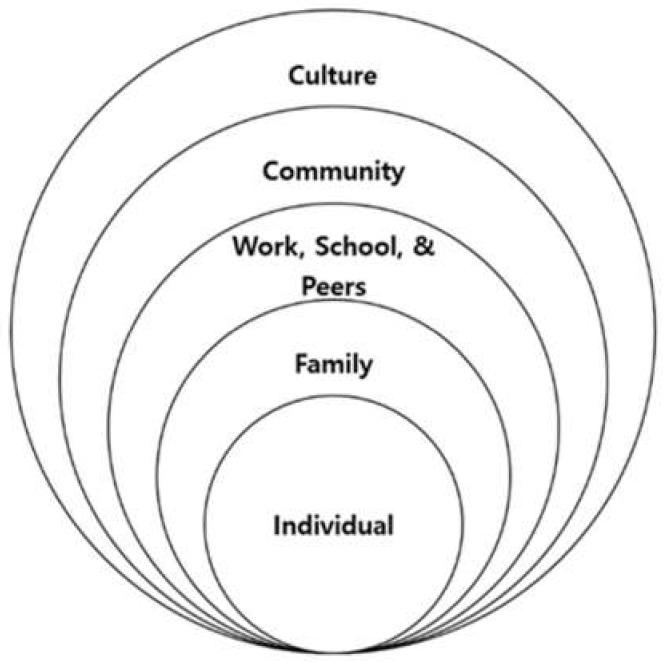
Modified socioecological systems theory (adapted from Bronfenbrenner, 1994 [10]).

**Table 1 ijerph-17-07917-t001:** Descriptive statistics of the study sample.

Variables	% (N), Mean (SD)
Flourishing (DV)	4.15 (0.69)
**1. Individual**	
Age	13.53 (2.35)
Female	48.72% (21,658)
Race/ethnicity:	
Non-Hispanic white	53.89% (30,496)
Non-Hispanic black	14.00% (4242)
Hispanic	20.77% (5216)
Other	9.01% (4368)
After-school activity (days)	1.56 (0.10)
Physical activity (days)	4.16 (2.32)
Sedentary behavior (hours)	4.04 (3.51)
Overweight	29.84% (12,788)
**2. Parenting capacity**	
Parent’s health (degree)	6.75 (2.40)
Stress-coping (degree)	3.53 (0.58)
Smoking	24.51% (10,580)
Alcohol and drug use	14.32% (6247)
Parent and child relationship (degree)	3.62 (0.61)
**3. Family function**	
Family activity (frequency)	6.84 (2.56)
Federal poverty level (FPL):	
0–99% FPL	19.87% (6013)
100–199% FPL	21.12% (7747)
200–399% FPL	28.88% (14,052)
400% FPL	30.13% (17,497)
Combined family structure and marital/cohabitation status:	
Single mother	19.44% (7353)
Others	79.26% (37,416)

*N* = 45,309; weighted N = 30,965,078; DV, dependent variable; FPL, federal poverty level. Data source: 2011–2012 National Survey of Children’s Health [11].

**Table 2 ijerph-17-07917-t002:** Sequential regression of the children’s flourishing with covariate.

Covariates.	Coef.	95% CI	Coef.	95% CI	Coef.	95% CI
1. Individual						
Age	0.00	(−0.01, 0.00)	0.00	(0.00, 0.01)	0.00	(0.00, 0.01)
Female	0.21 *	(0.18, 0.24)	0.20 *	(0.17, 0.22)	0.19 *	(0.17, 0.22)
Race/ethnicity:						
Non-Hispanic white	---		---		---	
Non-Hispanic black	0.00	(−0.04, 0.04)	−0.01	(−0.05, 0.03)	−0.02	(−0.06, 0.03)
Hispanic	0.13 *	(0.01, 0.19)	0.15 *	(0.10, 0.20)	0.14 *	(0.09, 0.19)
Other	0.07 *	(0.02, 0.11)	0.09 *	(0.05, 0.13)	0.09 *	(0.04, 0.13)
After-school activity	0.11 *	(0.09, 0.13)	0.09 *	(0.07, 0.10)	0.09 *	(0.07, 0.10)
Physical activity	0.03 *	(0.02, 0.04)	0.02 *	(0.01, 0.03)	0.02 *	(0.01 0.03)
Sedentary behavior	−0.03 *	(−0.03, −0.02)	−0.02 *	(−0.02, −0.01)	−0.01 *	(−0.02, −0.01)
Overweight	−0.02	(−0.05, −0.01)	−0.01	(−0.04, 0.03)	−0.01	(−0.05, 0.02)
**2. Parenting capacity**						
Parent’s health		0.02 *	(0.01, 0.03)	(0.01, 0.03)	0.02 *	(0.01, 0.03)
Stress coping		0.20 *	(0.16, 0.23)	(0.16, 0.23)	0.19 *	(0.15, 0.22)
Smoking		−0.02	(−0.05, 0.01)	(−0.05, 0.01)	−0.02	(−0.05, 0.02)
Alcohol and drug use		−0.15 *	(−0.19, −0.11)	(−0.19, −0.11)	−0.15 *	(−0.19, −0.11)
Parent and child relationship		0.31 *	(0.28, 0.34)	(0.28, 0.34)	0.30 *	(0.27, 0.33)
**3. Family function**						
Family activity					0.02 *	(0.01, 0.03)
Federal poverty level (FPL):						
0–99% FPL					--	
100–199% FPL					−0.01	(−0.06, 0.06)
200–399% FPL					−0.03	(−0.09, 0.03)
400% FPL					−0.01	(−0.07, 0.05)
Single parent					0.01	(−0.04, 0.07)
Wald tests			F (5,90) * = 159.05 (*p* < 0.01)		F (5, 90) * = 7.77 (*p* < 0.01)	

** p* < 0.01; *N* = 45,309; weighted N = 30,965,078; Coef.: coefficient; CI, confidence interval; Data source: 2011–2012 National Survey of Children’s Health [11].

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
