# Peer review of "Socioecological Predictors on Psychological Flourishing in the US Adolescence"

_ijerph, 2020, doi:10.3390/ijerph17217917_

Round 1

Reviewer 1 Report

0

Author Response

Thank you for your valuable review.

Reviewer 2 Report

No more comments

Author Response

Thank you for your valuable review.

Reviewer 3 Report

There are two issues, which need to be improved in the article. The first of these is that the introduction provides research,
which substantiates the work, but later, in discussion,
new research is used, which, previously, in the introduction,
has not served as a foundation, that is, the investigations
discuss whether we have served as a foundation. On the other hand, in the method section,
the type of research, the methodology used,
the research method and the type of sampling used have not been
specified.

Author Response

Thank you for your valuable review. We really appreciate the great comments and suggestions for the better research. Based on your comments, all changes are provided by a point-by-point response and also rewritten with red color in the manuscript. Please see the attachment.

This manuscript is a resubmission of an earlier submission. The following is a list of the peer review reports and author responses from that submission.

Round 1

Reviewer 1 Report

Where the study took place?

Could you give more information about the non-respondent distribution in each ethnic category?

I suggest presenting missing data for each variable reported.

Could you explain or hipotethize  about the lack of influence of family structure and socio-economical status? Is an non-intiutive result of your analysis.

Reviewer 2 Report

The publication "Socioecological predictors in psychological. Flourish in adolescence" presents a structure according to an exhaustive investigation, with a large sample and a correct methodological approach of interest for scientific knowledge. It delves into factors that will benefit the health and well-being of the family and the development of children and contributes to a new field of knowledge. The results are of interest and respond to the objectives set and a wording is shown in accordance with the requirements of the scientific documentation.

Reviewer 3 Report

When I first examined the content of this article entitled "Socioecological Predictors on PsychologicalIng in Adolescence", I found Sequential Regression of the Children’s Flourishing in the table 2made a serious error and therefore stopped the review. In Table 2, the significantness of the estimated parameters is not consistent with the results of 95% CI (confidence interval), indicating that there is an error in inferring the significantness of these coefficients. In addition, it is recommended that you discuss the correlation between variables and suggest listing the results of the "Correlations Matrix" before establishing the multiple regression models. Finally, it is recommended to exclude the existence of a multicollinearity in the models (table 2) established, and to use the variance inflation factor for testing. Since this article is submitted in "International Journal of Environmental Research and Public Health (ISSN 1660-4601; ISSN 1661-7827) ". This is a world-class, top journal and is included in the Journal of High Visibility in the Science Citation Index and Social Sciences Citation Index. Therefore, the internal text published is advanced and innovative, as well as a certain degree of contribution. For these reasons, I propose to temporarily refuse to publish in "International Journal of Environmental Research and Public Health ( ISSN 1660-4601; ISSN 1661-7827) ", and recommends that the author re-submit for retrial after revising the content and clarifying the empirical results of this article.